# Advancements in Standardizing Radiological Reports: A Comprehensive Review

**DOI:** 10.3390/medicina59091679

**Published:** 2023-09-17

**Authors:** Filippo Pesapane, Priyan Tantrige, Paolo De Marco, Serena Carriero, Fabio Zugni, Luca Nicosia, Anna Carla Bozzini, Anna Rotili, Antuono Latronico, Francesca Abbate, Daniela Origgi, Sonia Santicchia, Giuseppe Petralia, Gianpaolo Carrafiello, Enrico Cassano

**Affiliations:** 1Breast Imaging Division, IEO European Institute of Oncology IRCCS, 20141 Milan, Italy; luca.nicosia@ieo.it (L.N.); anna.bozzini@ieo.it (A.C.B.); anna.rotili@ieo.it (A.R.); francesca.abbate@ieo.it (F.A.); enrico.cassano@ieo.it (E.C.); 2Department of Radiology, King’s College Hospital NHS Foundation Trust, London SE5 9RS, UK; p.tantrige@nhs.net; 3Medical Physics Unit, IEO European Institute of Oncology IRCCS, 20141 Milan, Italy; paolo.demarco@ieo.it (P.D.M.); daniela.origgi@ieo.it (D.O.); 4Postgraduate School of Radiodiagnostics, University of Milan, 20122 Milan, Italy; serena.carriero@unimi.it; 5Division of Radiology, IEO European Institute of Oncology IRCCS, 20141 Milan, Italy; fabio.zugni@ieo.it (F.Z.); giuseppe.petralia@ieo.it (G.P.); 6Foundation IRCCS Cà Granda-Ospedale Maggiore Policlinico, 20122 Milan, Italy; sonia.santicchia@policlinico.mi.it (S.S.); gianpaolo.carrafiello@unimi.it (G.C.); 7Department of Oncology and Hemato-Oncology, University of Milan, 20122 Milan, Italy

**Keywords:** standardization, radiology report, radiology, structured report, AI, radiomics

## Abstract

Standardized radiological reports stimulate debate in the medical imaging field. This review paper explores the advantages and challenges of standardized reporting. Standardized reporting can offer improved clarity and efficiency of communication among radiologists and the multidisciplinary team. However, challenges include limited flexibility, initially increased time and effort, and potential user experience issues. The efforts toward standardization are examined, encompassing the establishment of reporting templates, use of common imaging lexicons, and integration of clinical decision support tools. Recent technological advancements, including multimedia-enhanced reporting and AI-driven solutions, are discussed for their potential to improve the standardization process. Organizations such as the ACR, ESUR, RSNA, and ESR have developed standardized reporting systems, templates, and platforms to promote uniformity and collaboration. However, challenges remain in terms of workflow adjustments, language and format variability, and the need for validation. The review concludes by presenting a set of ten essential rules for creating standardized radiology reports, emphasizing clarity, consistency, and adherence to structured formats.

## 1. Introduction

Medical imaging has revolutionized healthcare by providing non-invasive imaging techniques that help in diagnosis, treatment planning, and monitoring of diseases. Imaging is interpreted by clinical radiologists and the findings are communicated through written reports. Clarity and unambiguity of radiological reports are pivotal in translating imaging findings into actionable clinical information for physicians and other healthcare providers [1,2].

The implementation of standardized radiological reports has garnered significant interest and debate within the medical imaging field. Standardization refers to the process of transforming data into a common format that can be understood across different tools and methodologies [2,3,4]. Reporting templates have long been promoted by radiological societies to improve the completeness of reports and information localization by radiological and clinical colleagues for the purposes of direct patient care and data mining for research [1,2,5,6,7,8,9].

By establishing uniform reporting formats, standardized radiological reports offer several potential advantages. However, they also present challenges that need to be addressed. This review paper examines the implementation and impact of standardized radiological reports. It discusses the main advantages and disadvantages of standardized reporting, explores the efforts toward standardization, highlights recent technological advancements, and outlines the challenges associated with standardized reporting. The paper also discusses the potential role of AI in supporting the standardization of radiology reports and its impact on clinical practice.

## 2. Advantages and Disadvantages of Standardized Radiological Reports

Standardization aims to enhance the consistency and reliability of data input for radiomics and AI applications. By establishing uniform reporting formats, standardized radiological reports offer potential advantages as discussed below but it also presents challenges.

Clinger et al. [10] reported that 32% of clinicians preferred summary statements at the beginning of the report; meaning that some clinicians opt for a short and direct report in clinical practice. In contrast, Schwartz et al. [11] found better content and clarity satisfaction rates from structured reports than free-form, with a statistically significant difference. Standardized reporting offers multiple advantages over traditional free-text reporting, including improved clarity, increased efficiency as some reports can be generated more quickly [12], enhanced communication between radiologists [13], and checklists or templates being provided to help radiologists identify key findings and suggest appropriate follow-up actions [14]. Some studies have also found that standardized reports are easier for patients to read and understand [11,15].

If reports are not standardized, inconsistencies can lead to inaccurate diagnoses and limit research [2,16,17]. Standardization offers less room for ambiguity, which can reduce errors and variations in care. For example, a standardized report for mammography (MX) has been shown to improve the accuracy and completeness of reports, which can lead to earlier detection and treatment of breast cancer [18].

However, there are also potential disadvantages of standardized radiology reports. The constraints of option fields limit flexibility, and radiologists may be forced to choose from a limited set of pre-defined terms, which may not always communicate their interpretation [15,17]. This is potentially surmountable by increasing the number of options and permitting a free-text option that could be monitored to determine if it really is necessary. However, increasing options also increases the time and effort required to enter data into specific fields and adhere to a standardized format [19]. This is associated with a negative user experience, as radiologists, referring physicians, and patients may find it challenging to navigate standardized radiology reports, especially if they are not familiar with the format or language [20].

Recently, Rocha DM et al. [1] reviewed a wide range of literature to evaluate the main advantages and disadvantages of the structured radiological report. An analysis of 32 relevant publications showed that structured reports enhance clarity and readability, leading to improved communication and data quality. Structured reports also enhance the precision and accuracy of diagnostic information, making the data more legitimate and reliable. On the other hand, the review highlighted that structured reporting may be inadequate in complex cases due to oversimplification or inability to capture all necessary information. The authors also showed how resistance to change among professional radiologists is another barrier to the adoption of structured reporting, as it restricts the ability to write reports in their own voice. The main advantages and disadvantages of a standardized report are summarized in Table 1.

## 3. Efforts toward Standardization

Standardized reports have three main characteristics: a structured form with headings and paragraphs, a consistent organization with subheadings such as body parts, and a standard language [16]. All these attributes contribute to enhanced readability, navigation, and explicitly definitive communication of meaning among physicians. To reduce variations in terminology, the utilization of common and international imaging lexicons, such as the Fleischner Society glossary of terms, and the use of monosemous wording is required to provide a uniform method to share the information between radiologists, between radiologists and clinicians, and between radiologists and patients [1,2,11,21].

Several advancements have been incorporated into radiology reporting to support the formulation of standardized reports. An innovative approach to standardizing radiological reports involves the implementation of structured templates that guide radiologists in their reporting process and offer standardized terminology [13,22]. These templates utilize multiple-choice options, enabling the automatic generation of reports with reproducible terms. Such standardized reports are not only easily interpretable by healthcare professionals but can also be efficiently extracted and classified by software for further analysis. Figure 1 depicts an example of a standardized report developed at the European Institute of Oncology (Milan, Italy), illustrating the structure of the template and its various options. It is possible for some of the option fields to be provisionally completed by software solutions that automate the transfer of data from DICOM to reports, enhancing the user experience, report completeness, and expediting report generation [23].

Clinical decision support (CDS) tools based on current guidelines can be integrated into the radiologist’s workflow to input recommendations into the radiology report after the radiologist enters various data points. In a study, the implementation of a CDS tool was associated with improved adherence to Fleischer Society guidelines for incidental pulmonary nodules seen on CT, with significantly higher guideline concordance in cases where the tool was used (95%) compared to cases where it was not used (45%) [24].

Furthermore, recent technological advancements have resulted in a range of multimedia reporting solutions in radiology that enhance report communication and ease information extraction. Multimedia-enhanced reporting allows for interactive connections and hyperlinks to annotated images in PACS and provides radiologists with the ability to embed images, videos, tables, and graphs. Report links to annotated images enhance communication between radiologists by expediting lesion localization, and support clinicians. A specific example includes a urologist undertaking a targeted ultrasound-guided prostate biopsy based on a radiologist’s MRI interpretation and supported by the fusion of the annotated MRI sequences with the ultrasound images obtained during the biopsy procedure [25].

Other applications, such as semi-automated tumor measurement and tracking, offer an efficient and straightforward format for presenting longitudinal data. For instance, a graph of the Response Evaluation Criteria in Solid Tumors (RECIST) Index over time can provide clear and actionable information for the oncologist [26].

In oncology, with increasing multidisciplinary management, comprehensive and universally coherent radiological reports play a pivotal role. A growing amount of information is needed for multidisciplinary discussions and narrative reports have a high degree of variability and may miss essential treatment-choice information; therefore, they are not preferred in multidisciplinary case discussions [27]. Accordingly, in the last years, many organizations and initiatives have been established to standardize radiological reporting.

The American College of Radiology (ACR) in 1993 proposed the Breast Imaging Reporting and Data System (BI-RADS), which included recommendations for the conduct of mammograms (MX), containing an overall structure for MX reports and final assessment categories with management recommendations and a MX lexicon [28]. The standardization of radiology reports with the BI-RADS system has been of great benefit, aligning the language used by radiologists to clinical categorization and decision making. BI-RADS has allowed multi-institutional data collection, data sharing, auditing, and comparison. Since the success of BI-RADS, similar approaches have been taken for imaging other organs and body systems that provide a standardized framework for reporting on imaging findings and assessing the probability of disease. ACR reporting systems now include CT Colonography-RADS (C-RADS), Coronary Artery Disease-RADS (CAD-RADS), Head Injury Imaging-RADS (HI-RADS), Lung CT Screening-RADS (Lung-RADS), Neck Imaging-RADS (NI-RADS), Ovarian-Adnexal-RADS (O-RADS), Prostate Imaging-RADS (PI-RADS), Thyroid Imaging- RADS (TI-RADS), and Liver Imaging-RADS (LI-RADS). However, with few exceptions (e.g., PI-RADS [29] and O-RADS [30], which were established in collaboration with European societies), these projects were developed by US scientific societies only. BI-RADS, which is currently one of the most widespread examples of structured reports, evolved from a simple MX lexicon to a multimodality lexicon for MX, ultrasound (US), and magnetic resonance imaging (MRI), reducing variability and improving the clarity of communication between breast radiologists and other physicians [31]. The exhaustiveness of breast reports is the key point for the precision of the information in breast cancer care. Snoek et al. [32] demonstrated that structured radiology report in breast imaging has led to more complete pathology reports.

The European Society of Urogenital Radiology (ESUR) in 2012 published the first version of PI-RADS that included clinical indications for prostate MRI, minimal and optimal imaging acquisition protocols, and a structured category assessment system [33]. In 2015, the ACR, ESUR, and AdMeTech Foundation updated and improved the original version of PI-RADS and established a single international standard, PI-RADS Version-2 (PI-RADS v2), that overcomes the limitations of PI-RADS v1 [34,35,36].

There are different scoring systems for thyroid nodule classification. The most widespread are ACR-TIRADS and EU-TIRADS. Both classification systems have been demonstrated to be accurate methods to stratify thyroid nodules with high reproducibility [37].

Recently, the Radiological Society of North America (RSNA) developed the RadReport reporting templates to promote standardization in radiology reports. The online library of radiology reports [5] contains more than 250 report templates, searchable by keywords, specialty, popularity, language, or participating organizations [5]. Moreover, the RSNA also developed a standardized reporting lexicon called RadLex that can be used to describe radiological findings, leading to better communication and consistency in reporting [38].

The European Society of Radiology (ESR) and the RSNA have established the Template Library Advisory Board and have opened a platform where all users, not only RSNA members, can upload and discuss their report templates [5].

Various consensus statements have been published by other societies providing disease-specific report templates, which are shown to provide the most benefit to referring physicians. The American Society of Abdominal Radiology published a statement regarding structured reports for pancreatic ductal adenocarcinoma [39], and the Korean Society of Abdominal Radiology published a statement regarding structured reports regarding cancer MRI [8].

The ESR has developed the EuroSafe Imaging initiative, which aims to promote radiation protection, including the standardization of reporting [40].

In 2013, the ACR with the RSNA introduced the Imaging 3.0 initiative to shepherd the shift from volume-based care to value-based care in radiology [9]. One mechanism through which radiology is achieving this paradigm change is through improvements to the major work product of the radiology imaging value chain: the radiology report. In the Imaging 3.0 initiative, informatics committees are collaborating toward shaping the direction of template development and facilitating the inclusion of specified findings and recommendations.

## 4. Challenges in Standardized Radiological Reports

Implementing standardized solutions comes with challenges, such as the requirement for custom mapping and the arduous task of testing, validating, and debugging metrics [23].

The standardization of radiological reports requires a significant effort from radiologists, healthcare providers, and software developers. The implementation of standardized reports may require changes to the workflow and reporting systems of healthcare providers, which can be time-consuming and costly. Furthermore, the standardized output may not be universal. For example, a template report structured toward local reimbursement may be aligned with a unit’s financial strategy but be non-contributory toward AI usable data [41].

The Royal College of Radiologists (RCR) produced guidelines on reporting standards and clinical radiology workloads [42]. These guidelines are used to assess the quality of radiological reports and in workforce planning and reviews. The guidelines recommend the use of standard terminology suitable for data collection; however, these may not be explicitly AI-oriented. Additionally, changes to radiological reporting to meet the needs of AI may fall outside the current guidelines and would need to be validated in the research setting prior to their implementation in nationally funded clinical units.

Despite the availability of recommended terms for radiological reporting, the standardization of radiological reports in language, style, and format remains a challenge. Radiologists often use different language to describe the same findings, leading to misinterpretation by other healthcare providers. The use of abbreviations, eponyms, and acronyms can also create confusion and misunderstandings. In addition, radiologists have different reporting styles, which can lead to inconsistencies in the report’s structure and content. The lack of standardization in the format of the report can make it challenging to compare reports during longitudinal follow-up and across different healthcare systems. Therefore, the first step to reach a standardized report is to reduce variations in terminology in medical reports through the adoption of a common and international imaging lexicon for improved communication between different medical specialties and centers [1,2,11,17,21,43].

To ensure clarity, templates should be concise, and excess text, such as redundant descriptions of acquisition techniques in the header, should be minimized [2,22]. Templates should reflect a natural flow for both the interpreting radiologists and referring physicians. For example, body CT reports for scans obtained for a suspected malignancy of unknown origin could describe organ findings in a cranial to caudal order, while brain magnetic resonance imaging reports could describe intra-axial, extra-axial, and extracranial pathology in order.

Furthermore, both clarity and consistency can be further improved if residents are taught to avoid pitfalls, such as using different templates for a single indication (e.g., Facial CT and Sinus CT when reporting facial pain), using redundant fields (e.g., upper abdomen and adrenals as two separate fields in a chest CT), and using advanced formatting like bullet lists and indentations, which may not convert correctly between dictation software and electronic medical record user interfaces [22].

From the workflow perspective, it is favorable to produce the radiological report while evaluating the acquired images, without needing to divert attention from the imaging screen to a report screen. Voice recognition tools have addressed this need and could be incorporated into template reports. The initial challenge for AI would be to enable the radiologist to produce a standardized report through uninterrupted dictation. The AI could convert the radiologist’s input into a desired format, or multiple desired formats depending on needs, and extract the data required for itself to develop and enhance AI tools.

## 5. AI Can Help the Standardization of Report

While much attention has been focused on AI’s role in interpreting and classifying imaging findings [44], AI may also aid in the overall delivery of radiological services including radiology reporting systems, potentially automating certain parts of report generation [45]. For instance, a simple AI algorithm could extract the date of a comparison study and insert it into the report, improving report comprehensiveness. AI algorithms could also improve the accuracy of reporting and data systems by determining if an appropriate comparison study is available and automatically inserting whether threshold growth of a selected lesion is present [45]. Similarly, AI could auto-insert recommendations for additional testing or follow-up imaging if critical results or the need for non-urgent follow-up are recognized within reports [46]. AI could also generate multiple versions of the radiology report customized for different stakeholders, improving communication [47,48].

Nowadays, AI software can automatically generate radiology reports applicable to X-rays [49]. The automated generation of radiology reports is available for X-rays and has tremendous potential to enhance patients’ clinical diagnosis of diseases. For example, CADxReport is a learning-based technique for generating clinically accurate reports from chest X-ray (CXR) images that generate sufficiently accurate CXR reports [50]. Paalvast et al. [51] show that the quality of the generated reports approximates the quality of the original reports and highlights challenges in creating sufficiently detailed and versatile training data for automatic radiology report generation. The experimental results conducted on the Indiana University Chest X-ray datasets and MIMIC datasets demonstrate that the proposed model of automatic report generation achieves state-of-the-art performance compared with other baseline approaches [49,52].

## 6. Methods for Structuring Reports from Unstructured Reports

Unlike standardized reporting, the definition of structured reporting is less clear in the current literature [19]. However, there is a consensus about the concept that structured reports must help the writer create their report, through either a predefined design, template, or a checklist and, at the same time, should help readers to understand the main message of the report [2].

Today, most of the radiology reports are neither structured nor standardized [53]. To address the challenge of standardizing free-diction radiological reports, natural language processing (NLP) techniques can be used. NLP techniques involve analyzing and interpreting natural language data, such as free text radiology reports. By structuring the reports, the data can be used for analysis and research.

The first step in structuring unstructured reports is to identify the relevant information in the report. This involves extracting information such as patient demographics, clinical history, imaging findings, and impressions. NLP techniques can be used to identify the relevant information by analyzing the text and identifying patterns and keywords. Once the relevant information has been identified, it can be structured using a standardized reporting format. The ESR has developed reporting templates, guidelines, and ontologies to ensure uniformity in reporting. These standardized reporting formats can be used to structure the extracted information, leading to standardized and consistent reports (Figure 2).

Several studies have shown the ability of NLP networks to extract information that can be presented in structured form [54,55]; however, they mostly dealt with retrospective automated data analysis. The work of Jorg et al. [56] is the first to use NLP for the generation of structured reports, as well as the first approach to integrating a dialogue system into the reporting process. Creating structured reports from free-form dictation could have a huge impact on clinical workflow, as radiologists can continue to rely on the speech recognition that they have been conveniently using for decades, while also benefiting from the above-mentioned advantages of standardized reports.

## 7. 10 Rules to Create a Standardized Report

Standardized radiology reports play a critical role in improving communication, enhancing patient care, and facilitating research and data analysis in the field of radiology. To ensure the utmost utility and effectiveness of these reports, radiologists must adhere to specific guidelines and principles. In this section, we present ten essential rules that radiologists should follow when creating standardized radiology reports. These rules are designed to enhance clarity, consistency, and the overall quality of the reports, ultimately leading to better patient outcomes and increased collaboration among healthcare professionals.

Be Clear and Concise

Ensure that the report conveys the findings and interpretations in a clear and succinct manner. Avoid unnecessary technical jargon and use straightforward language that can be easily understood by referring physicians and other healthcare providers.

2.Use Structured Templates

Organize the report using a structured format that includes clear sections for clinical information, imaging techniques and lesion description. Adopt structured reporting templates that are specific to different imaging modalities and clinical scenarios. These templates help organize information systematically and provide a standardized framework for reporting findings.

3.Include Relevant Clinical Information

Incorporate pertinent clinical history and indications for the examination. This context assists in accurate interpretation and ensures that the report aligns with the clinical question being addressed.

4.Provide Contextual Recommendations

Offer appropriate recommendations based on the radiological findings. This could include further imaging studies, additional diagnostic tests, or consultations with other specialists.

5.Follow Consistent Nomenclature:

Use standardized and well-defined medical terminology to describe imaging findings, ensuring uniformity across reports. Employ established lexicons to ensure accurate and consistent terminology usage. Comprehensive Lesion Description: Provide a detailed description of the lesion(s) of interest, including size, location, shape, margins, and internal characteristics. Standardized descriptors, such as BI-RADS for breast imaging [28], can be utilized to ensure consistent reporting across different institutions.

6.Utilize Imaging Protocols:

Adopt and adhere to standardized imaging protocols to ensure uniformity and consistency in image acquisition. Consistent imaging protocols reduce variability in the reported findings and aid in comparing results over time.

7.Incorporate Structured Reporting Elements:

Include specific elements in the report, such as measurements, location descriptors, and imaging characteristics, in a structured format. This enhances the completeness of the report and facilitates data mining and research.

8.Address Critical Findings Promptly:

For critical or urgent findings, ensure that the report communicates the information immediately to the referring physician or the appropriate healthcare team. Timely communication is vital for patient management and safety.

9.Validate and Review Reports:

Establish a process for validation and peer review of standardized reports to maintain accuracy and quality. Regular feedback and validation help identify areas for improvement and ensure adherence to the defined standards.

10.Update and Evolve:

Regularly update the standardized reporting templates and guidelines based on emerging evidence, technological advancements, and feedback from users. Continuous improvement ensures that the reports remain relevant and effective over time.

## 8. Conclusions

In conclusion, standardized radiological reports represent a crucial step forward in optimizing healthcare outcomes through effective communication and improved data quality. While standardized reporting offers numerous advantages, it is important to acknowledge the challenges it poses, including potential limitations in capturing complex cases and user resistance to change. Efforts toward standardization, driven by radiological societies, academic institutions, and technological advancements, are instrumental in shaping the future of medical imaging reporting. The integration of AI technologies shows promise in streamlining the reporting process and enhancing its accuracy. As the field of radiology continues to evolve, ongoing collaboration, feedback, and innovation are vital to ensuring the continued refinement and effectiveness of standardized radiological reports.

## Figures and Tables

**Figure 1 medicina-59-01679-f001:**
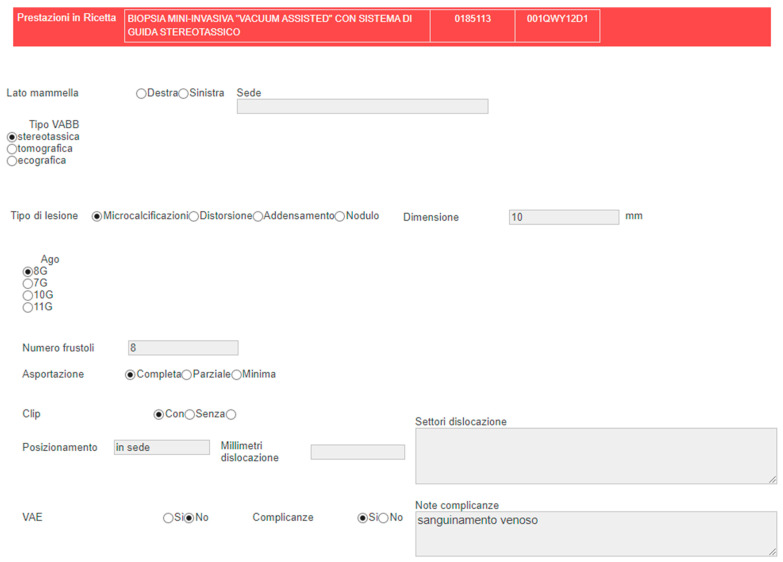
A standardized radiological report template developed at the University Hospital IEO (European Institute of Oncology) for a reporting an imaging-guided vacuum-assisted breast biopsy. The template consists of clearly defined sections, including clinical information, imaging techniques, type of procedure, needle details, and lesion description. The radiologist is guided through the reporting process with multiple-choice options, ensuring consistency in data input. The resulting report contains reproducible terms and can be easily extracted and classified by software for subsequent radiomics and AI analysis.

**Figure 2 medicina-59-01679-f002:**
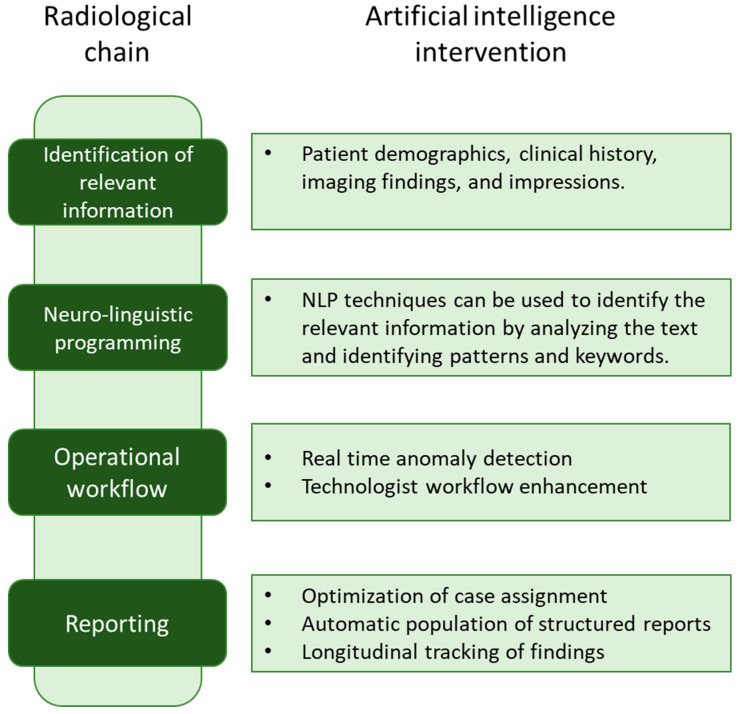
The process to structure unstructured reports. Natural language processing (NLP) techniques extract relevant information from the report, including patient demographics, clinical history, imaging findings, and impressions. The extracted details are then organized using standardized reporting formats, promoting consistency and uniformity in the reports.

**Table 1 medicina-59-01679-t001:** Advantages and disadvantages of standardized radiological reports.

Advantages	Disadvantages
Improved clarity and completeness Increased efficiency Improved communication Facilitates decision-making	Limited flexibility Increased time and effort Inconsistencies in interpretation Poor user experience

## Data Availability

Not applicable.

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
