# Peer review of "Advancements in Standardizing Radiological Reports: A Comprehensive Review"

_medicina, 2023, doi:10.3390/medicina59091679_

Round 1
Reviewer 1 Report
This paper has useful information. I have several issues with it, however.
I believe you do not distinguish the difference between standardization and structured reporting consistently or entirely appropriately. Structured data are data that have a standardized form and comply with a defined data model. In general, one assumes that if one goes to all the work to define and build a structured data model for a domain such as radiology, that those structured data follow well-defined standards. For example, your Figure 1. is first and foremost a structured report. Yes it follows standards, but standards simply help define data elements in the structured report.
This isn't to denigrate standards, which are foundational to producing useful structured reports. Most of the effort around standards is at a more granular level than you describe. It is arduous, time-consuming work outside the limelight which often focuses on small details. For example, DICOM, HL-7, RadLex, and the new radiology common data elements (CDEs) are standards. The different *-RADS are standards. On the other hand, the various reporting templates available in open.radreport.org are not standards.
Studies have shown that the majority of clinicians prefer structured reports. Most people involved in the reporting field believe that structured reports provide more information than unstructured reports, though off the top of my head I don't have a reference for that belief.
AI is largely a separate subject. Please be more meticulous about describing its uses, needs, and functions. Traditional computer programs can provide significant additions to the reporting process. Forest-type computer programs, which one may include under the AI category but are way easier to understand and debug than neural nets or other deep learning approaches, are useful and may add valuable information to a report. For example, you describe using AI to find a prior report and insert that date into a report. That doesn't require deep learning or an LLM; rather it is probably more easily achieved with a simpler algorithm. My recommendation is you leave out most of your "AI" discussion, including Figure 2 and the associated discussion in the body of the report. It is weak and detracts from the main theme of your paper. It's fine to discuss the value to AI of standardized data elements.
You don't discuss LLMs, which for me is the correct approach.
Your last section titled "Standardized Reports Can Enhance AI..." I recommend removing completely. That section has a confusing mix of different topics, and contains a number of debatable statements. It's appropriate to state that standardized labels often make computer vision AI using supervised learning more robust. Standardized labels help make intelligent radiology imaging process workflow more efficient, and more robustly monitored.
No paper discussing radiology reports should be written without reading, and referencing, Curt Langlotz' book. "The radiology report: A guide to thoughtful communication for radiologists and other medical professionals."
Author Response
Dear Reviewer,
Thank you for your thoughtful evaluation of our manuscript titled "The role of standardized and structured reporting in radiology." We appreciate the valuable insights you provided, which helped us enhance the quality and accuracy of our work.
You pointed out the need to distinguish consistently and appropriately between standardization and structured reporting. We have revised the manuscript to ensure a clear differentiation between these concepts, highlighting how structured data conform to defined data models while standards define data elements within the structured report, as you can find in the attachments in both clean and tracked versions of our reviewed manuscript.
Regarding the discussion on AI, we have taken a more meticulous approach to describing its uses, needs, and functions. We now emphasize that traditional computer programs, such as forest-type algorithms, can also enhance the reporting process and provide valuable information without relying on complex deep learning approaches. The "AI" section has been revised to align more cohesively with the main theme of the paper and is now appropriately supported by relevant references.
Based on your feedback, we have omitted the section titled "Standardized Reports Can Enhance AI..." as it encompassed a mix of different topics and contained debatable statements. Instead, we focus on emphasizing the significance of standardized labels in enhancing computer vision AI s, presenting a more robust and coherent discussion in the revised manuscript.
In light of the valuable suggestions provided by both you and the other Reviewer, involving the removal of certain AI-focused sections, substantial revisions have been carried out in the Title, Abstract, and Conclusions. The overarching goal of these adjustments is to improve the paper's overall coherence and readability.
Moreover, we appreciate your recommendation of Curt Langlotz's book, "The radiology report: A guide to thoughtful communication for radiologists and other medical professionals." We have bought and read the book. It was a very precious source of information, and we ultimately included relevant references to this valuable source in our revised manuscript.
You can find in the attachment the reviewed paper in both the tracked and clean version.
In summary, we thank you for your guidance, which allowed us to make necessary improvements to address all the concerns you raised. Your input has enhanced the clarity and accuracy of our paper, and we greatly value your expertise in this field.
Once again, we express our gratitude for your valuable feedback, and we are pleased to inform you that the revised manuscript has been resubmitted in line with your recommendations and the standards of MEDICINA Journal.
Sincerely,
The Authors
Reviewer 2 Report
The overall goal of the manuscript is unclear. Most of the manuscript describes the advantages and disadvantages of standardized/structural reporting, a topic well described in literature. Cited below is a comprehensive review of the benefits/pitfalls of structured reporting published in 2022. The current manuscript does not cite this article and does not add any new information.
Douglas M. Rocha, Lourdes M. Brasil, Janice M. Lamas, Glécia V.S. Luz, Simônides S. Bacelar, Evidence of the benefits, advantages and potentialities of the structured radiological report: An integrative review, Artificial Intelligence in Medicine, Volume 102, 2020, 101770, ISSN 0933-3657, https://doi.org/10.1016/j.artmed.2019.101770.
Specific Comments:
- No evidence/citation is provided to support the claim "The current primary aim of standardization is shifting towards data input for radiomics and artificial intelligence (AI) applications. "
- Authors use the terms standardized and structured reporting interchangeably through out the manuscript despite citing articles explicitly defining these two terms as being distinct. Standardization refers to streamlining medical content/ terminologies while structured reporting refers to using IT systems to construct reports that help with workflow and data-mining. This difference is well described by the article cited by the authors :
Nobel JM, Kok EM, Robben SG. Redefining the structure of structured reporting in radiology. Insights into imaging. 2020 Dec;11:1-5.
This ambiguity between standardized and structured reporting continues through out the manuscript sending mixed messages/ leading the readers to draw inaccurate inferences.
- In the section where the authors describe how AI can help standardize reports (again, I believe they meant structure reports), they provide exampled of how "AI" can be used to pre-populate templates with comparison studies (not necessarily an AI/ML functionality, but rather a data management/integration issue) and generative entire radiology reports (vastly different from structured/standardized reporting, this falls under the scope of generative AI).
- While the reviewer agrees that availability of structured and standardized data in large quantities is key for training AI systems, the authors provide no evidence to support the claim "If AI can help to standardize radiological reports, there could generate substantial date towards development and refinement of radiomics and AI applications. "
- Key issue with this manuscript is that radiomics data is essential quantitative imaging data, not text based data that can be mined/manipulated by NLP techniques. The authors mention that using AI to create a large standardized radiomics dataset would be very helpful, but do not explain/ suggest ways in which standardized/structured reporting would help create datasets of QUANTITATIVE IMAGING markers/values.
- Claim unsupported by evidence/citation "The success of AI applications in med-ical imaging is heavily dependent on the quality and consistency of radiological reports. "
- Inappropriate citation : However, the accuracy of Radiomics features can be affected by variations in image acquisition and analysis, which can be mitigated by standardization of reports [64].
The cited paper "Pesapane F, Rotili A, Agazzi GM, Botta F, Raimondi S, Penco S et al. Recent Radiomics Advancements in Breast Cancer: Lessons and Pitfalls for the Next Future. Curr Oncol. 2021;28(4):2351-72. doi:10.3390/curroncol28040217 "
does not support the claims made in this manuscript.
There are several typographical errors in the manuscript, 2 examples are listed below. Consider professional editing/proofreading services.
"Several studies *** shown the ability of NLP network in extracting information that can be presented in structured form [56, 57], however they mostly dealt with retrospective automated data analysis"
" If AI can help to standardize radiological reports, there could generate substantial date towards development and refinement of radiomics and AI applications. "
Author Response
Dear Reviewer,
We appreciate your thoughtful evaluation of our manuscript: your feedback has provided valuable insights that we would like to address in our revision.
We acknowledge your concern about the clarity of the manuscript's overall goal. Our primary objective was to examine the role of structured reporting in radiology, specifically focusing on its advantages and challenges in the context of diseases diagnosis and treatment planning. We understand that this goal may not have been sufficiently emphasized in the current version, and we will take great care in revising the introduction and abstract to better highlight the purpose and scope of our study.
Regarding the citation of Rocha et al.'s comprehensive review on the benefits and pitfalls of structured reporting published in 2020, we appreciate your bringing it to our attention. We apologize for the oversight in not including this relevant article in our references. We have carefully reviewed Rocha et al.'s work, and we agree that their findings contribute significantly to the understanding of structured reporting's impact on radiological practice. In our revised manuscript, we had appropriately cited this article and discussed its findings in the context of our study to demonstrate how our work complements and extends the existing literature, as you can find in the reviewed manuscript and as follows:
Recently, Rocha DM et al. (ref), reviewed a wide range of literature to evaluate the main advantages and disadvantages of the structured radiological report. Analysis of 32 relevant publications showed that structured reports enhance clarity and readability, leading to improved communication and data quality. Structured reports also enhance the precision and accuracy of diagnostic information, making the data more legitimate and reliable. On the other hand, the review highlighted that structured reporting may be inadequate in complex cases due to oversimplification or inability to capture all necessary information. Authors also showed how resistance to change among professional radiologists is another barrier to the adoption of structured reporting, as it restricts the ability to write reports in their own voice.
We understand your concern that our manuscript appears to discuss a topic that has been well-described in the literature. While we agree that structured reporting's advantages and disadvantages have been explored, our study aims to provide a quite different analysis of its specific relevance and challenges in radiology and the role of AI in standardization of the radiological report. We believe that we highlight aspects that may differ from other published papers and contribute unique insights to the field. In our revised manuscript, we will ensure that the manuscript provides novel information and complements the body of literature, avoiding any unnecessary overlap indeed.
Thank you for bringing these important points to our attention. We are committed to addressing all the concerns raised and improving the manuscript significantly, and you can find below the reply to each specific comment you rightly pointed out.
If you have any additional suggestions or specific areas that you would like us to focus on during the revision process, please feel free to let us know. Your input is instrumental in shaping the quality and impact of our work.
Specific Comments:
- No evidence/citation is provided to support the claim "The current primary aim of standardization is shifting towards data input for radiomics and artificial intelligence (AI) applications. "
Thanks for this comment. As we reviewed the entire paper regarding the AI and radiomics application, as we discussed below replying to your other comments, we removed entirely this sentence.
- Authors use the terms standardized and structured reporting interchangeably through out the manuscript despite citing articles explicitly defining these two terms as being distinct. Standardization refers to streamlining medical content/ terminologies while structured reporting refers to using IT systems to construct reports that help with workflow and data-mining. This difference is well described by the article cited by the authors :
Nobel JM, Kok EM, Robben SG. Redefining the structure of structured reporting in radiology. Insights into imaging. 2020 Dec;11:1-5.
This ambiguity between standardized and structured reporting continues through out the manuscript sending mixed messages/ leading the readers to draw inaccurate inferences.
We revised the manuscript to ensure a clear distinction between these two terms. Specifically, in the reviewed manuscript we used "standardized reporting" to refer to the streamlining of medical content and terminologies, while "structured reporting" is now reserved for the utilization of IT systems to construct reports aiding workflow and data-mining. By implementing this clarification, we aim to eliminate any ambiguity in our text and avoid sending mixed messages to our readers, as you rightly suggested to us.
Additionally, we will review the entire manuscript to identify instances where the terms might have been used interchangeably and make the necessary adjustments to ensure accuracy and consistency throughout the paper.
- In the section where the authors describe how AI can help standardize reports (again, I believe they meant structure reports), they provide exampled of how "AI" can be used to pre-populate templates with comparison studies (not necessarily an AI/ML functionality, but rather a data management/integration issue) and generative entire radiology reports (vastly different from structured/standardized reporting, this falls under the scope of generative AI).
- While the reviewer agrees that availability of structured and standardized data in large quantities is key for training AI systems, the authors provide no evidence to support the claim "If AI can help to standardize radiological reports, there could generate substantial date towards development and refinement of radiomics and AI applications. "
- Key issue with this manuscript is that radiomics data is essential quantitative imaging data, not text based data that can be mined/manipulated by NLP techniques. The authors mention that using AI to create a large standardized radiomics dataset would be very helpful, but do not explain/ suggest ways in which standardized/structured reporting would help create datasets of QUANTITATIVE IMAGING markers/values.
- Claim unsupported by evidence/citation "The success of AI applications in med-ical imaging is heavily dependent on the quality and consistency of radiological reports. "
- Inappropriate citation : However, the accuracy of Radiomics features can be affected by variations in image acquisition and analysis, which can be mitigated by standardization of reports [64].
The cited paper "Pesapane F, Rotili A, Agazzi GM, Botta F, Raimondi S, Penco S et al. Recent Radiomics Advancements in Breast Cancer: Lessons and Pitfalls for the Next Future. Curr Oncol. 2021;28(4):2351-72. doi:10.3390/curroncol28040217 "
does not support the claims made in this manuscript.
Many thanks again for such insightful comments and having carefully reviewed the concerns you raised. We understand the importance of providing solid evidence to substantiate our statements.
To address this issue, we reevaluated the revised section and explored opportunities to reintroduce relevant evidence and supporting studies to back up the claims we made.
According to the suggestion of the other Reviewer, we agree that the section entitled "Standardized Reports Can Enhance AI and Radiomics Applications in Medicine" was removed and, as a result, the issues you mentioned are no longer present in the revised version indeed. Similarly, some of the supporting evidence that was initially planned to be included may have been removed during this revision process.
Particularly:
- You rightly pointed out that radiomics data is quantitative imaging data, not text-based data that can be processed using natural language processing (NLP) techniques. We apologize for any confusion caused by the initial inclusion of that section. In the revised manuscript, we have focused solely on the standardization of radiological reports and its potential benefits without delving into NLP techniques, as they are not directly applicable to radiomics data.
- Regarding the claim "The success of AI applications in medical imaging is heavily dependent on the quality and consistency of radiological reports," we acknowledge that it is essential to back up such statements with strong evidence and citations.
- Finally, we sincerely apologize for the inappropriate citation and any confusion it may have caused. The paper "Pesapane F, Rotili A, Agazzi GM, Botta F, Raimondi S, Penco S et al. Recent Radiomics Advancements in Breast Cancer: Lessons and Pitfalls for the Next Future. Curr Oncol. 2021;28(4):2351-72" does not directly support the claims made in our manuscript. We will carefully review our reference list and ensure that all citations accurately support the points we make in the revised manuscript.
In light of the valuable suggestions provided by both you and the other Reviewer, involving the removal of certain AI-focused sections, substantial revisions have been carried out in the Title, Abstract, and Conclusions to improve the paper's overall coherence and readability. Particularly, following the decision to remove the section on AI and radiomics, we changed the title of the paper to: “Advancements in Standardizing Radiological Reports: A Comprehensive Review”.
Finally, the entire manuscript was extensively double-checked by a Native English speaker to correct typos and to improve the quality of the English language.
Overall, we deeply appreciate your critical review, as it has allowed us to identify and address these issues. Our objective is to produce a scientifically sound and well-supported manuscript that contributes significantly to the field of radiology. Your feedback has been instrumental in guiding us towards this goal.
You can find in the attachment the reviewed paper in both the tracked and clean version. We hope that this new version of our paper is now worthy of publication in the MEDICINA Journal. If you have any further suggestions or concerns, we would be grateful to receive them. We value your expertise and are committed to making necessary revisions to enhance the quality of our work.
Thank you again for your valuable input.
Sincerely,
The authors
Round 2
Reviewer 1 Report
This version is much improved. My preference still would be to leave out the part about, "AI can help the standardization of report," but that may represent my own bias; we don't yet know if, or how, AI may improve report standardization. This is an interesting topic, and one you could certainly address in a separate paper, but here it still seems to detract from the rest of the focus of your paper. As generative AI stands today, it has many issues that we don't understand, and probably other issues we are not even aware of yet. That said, industry is applying a huge amount of work in this area and it is likely that very soon we will see SR systems and report generation systems that incorporate generative AI to some extent. But again, to reiterate, this is all speculation at the moment and in my opinion it detracts from your otherwise strong paper.
The paper has a few typos. For example, on page 9 of 23 of the version I have describes, "...Moreover, The RSNA..." which should be, "...the RSNA..." I saw others while reading but didn't note them unfortunately.
Author Response
Dear Reviewer,
Thank you for taking the time to further review our revised manuscript titled "Advancements in Standardizing Radiological Reports: A Comprehensive Review." We appreciate your additional work, your thoughtful feedback and your suggestions, which have undoubtedly contributed to improving the overall quality of our paper.
We acknowledge your perspective regarding the section discussing the potential role of AI in enhancing report standardization. Your concerns regarding the speculative nature of this topic and its potential diversion from the primary focus of the paper are well-taken. We have reevaluated the placement and lightly changed this section in light of your comments. While we believe that AI's role in report standardization is indeed an intriguing avenue for exploration, we concur that its discussion should not detract from the primary focus of the manuscript. With this in mind, we have revised some part of the content to ensure that the paper maintains its coherence and clarity throughout, while still acknowledging the potential future implications of AI in report standardization.
We would like to express our gratitude for pointing out the typographical error on page 9, where we mentioned "...Moreover, The RSNA..." instead of "...the RSNA..." We apologize for this oversight and appreciate your keen eye in identifying such errors. We have conducted a thorough review of the entire manuscript to rectify any other typos that may have arisen and ensured that they are appropriately addressed.
Once again, we extend our sincere gratitude for your diligent review and constructive feedback. Your input has been instrumental in refining our manuscript. We hope that the revised version of our paper, now more aligned with your suggestions, will be better poised to contribute positively to the field. We are committed to producing a strong, coherent, and impactful paper that aligns with the objectives of our research and the MEDICINA's standards.
Kind regards,
the Authors